# The Buckling and Post-Buckling of Steel C-Columns in Elevated Temperature

**DOI:** 10.3390/ma13010074

**Published:** 2019-12-22

**Authors:** Leszek Czechowski, Sławomir Kędziora, Zbigniew Kołakowski

**Affiliations:** 1Department of Strength of Materials, Lodz University of Technology, 90-924 Lodz, Poland; zbigniew.kolakowski@p.lodz.pl; 2Faculty of Science, Technology and Communication; Luxembourg University, L-1359 Luxembourg, Luxembourg; slawomir.kedziora@uni.lu

**Keywords:** thermal buckling, post-buckling state analysis, finite element method, Koiter’s theory, thin-walled structures

## Abstract

This work deals with the investigation of a steel thin-walled C-column subjected to compression due to temperature increase. These experimental studies of the compressed columns in post-buckling state were conducted to determine their load-carrying capacity. To ensure appropriate supports and keeping of columns, plates with grooves were constructed. The tests of the columns’ compression for different preloads were carried out. By comparing the experiment results, numerical calculations based on the finite element method (FEM) and the semi-analytical method (SAM) of solution were performed. The computations were executed with the use of full material characteristics with consideration of large strains and deflections. Furthermore, while observing the deformation of columns, a non-contact Digital Correlation ARAMIS^®^ system was employed whose calculated results of deformations are very close to the results of the numerical method. The paper revealed that maximum recorded loads under temperature rise are comparable regardless of a value of initial load. A good correlation in results between used methods was achieved. The main goal of the present work was to assess of behavior of thin-walled compressed steel columns in a temperature-controlled environment till their full damage.

## 1. Introduction

Low-carbon steel DC01 is a material commonly applied in industry to elementary elements for the purposes accessibility and lower price. Nowadays, many strength tests on materials must be performed within a temperature field because some behavior of structures in higher temperatures should be known at the design stage. Hence, it can be stated that the behavior of any structures under thermal load can differ from the behavior under mechanical loads, dependent upon work temperature as given in reference [1]. Researchers in other papers [2,3] focused on analysis of thermal buckling of circular plates. These works show the use of the Digital Image Correlation (DIC) technique to investigate the thermal buckling of circular laminated composite or aluminum plate subjected to a uniform distribution of temperature load. Nguyena et al. in their work [4] analyzed experimentally and numerically a carbon fiber reinforced polymer (CFRP) structure under elevated temperature and different mechanical loads. Authors in reference [5] investigated the mechanical properties of S30408 austenitic stainless steel in the temperature field to determine stress-strain curves in tension even in 900 °C. Fang and Chan in their paper [6] studied S460 steel columns under axial compression due to elevated temperature by the use of the finite element method. They also performed the compression of those columns experimentally but at the ambient temperature. Li and Young in reference [7] analyzed the samples built of cold-formed high-strength steel due to tension under temperature coming to 1000 °C based on existing standards. Sua et al. in reference [8] studied the behavior of aluminum alloy beams at elevated temperatures ranging from 24 °C to 600 °C. They validated the experiment employing the finite element method. Likely, the authors in paper [9] examined the bending of beams made of high-strength steel in temperatures of up to 1000 °C. On the other hand, taking into consideration of analyses of elements subjected to a temperature field in theoretical aspects, one can find many papers devoted to the thermal buckling and the thermal post-buckling of thin-walled structures made of functionally graded material in references [10,11,12,13]. Studies on the stability of structures under mechanical loads were widely included in papers [14,15,16,17,18,19], among others.

Based on allowable literature, only a few papers concerning an experimental study of the stability of thin-walled structure under elevated temperature can be found. As this revealed, the analysis of these structures in these conditions is still desirable. The present paper deals with the post-buckling state of thin-walled open columns under compression due to temperature rise. The edges of columns during tests were inserted into grooves of the plates to reflect the articulated supports often used in the literature [20,21,22]. The experimental studies were executed until the drop of the compression force (maximum value of the load is often called the load carrying capacity). The whole paths of equilibrium with different preloads were registered. Additionally, Digital Image Correlation ARMIS^®^ system (DICAS) assigned to a record of the map deformation was employed during the experiment. To validate empirical results, numerical simulations based on a finite element method and semi-analytical methods (SAM) grounded on asymptotic Koiter’s theory [23] were carried out. In numerical computations, the full characteristic of steel attained due to one-directional tension at the ambient temperature was taken into account. The results of investigations were analyzed with regard to thermal post-buckling behavior of the columns. It was revealed that initial loads have an essential influence on the equilibrium paths.

## 2. Problem Description

### 2.1. Object of Investigation

The thin-walled C-profile column with dimensions of L = 250 mm, b = 80 mm, a = 40 mm, and thickness t = 1 mm was subjected to compression due to temperature rise, which was taken into consideration (Figure 1). The inserted columns between plates were initially compressed and afterwards heated. The material of analyzed columns was cold-rolled steel (symbol DC01) according to *EN 10139* standard at composition of elements: C = 0.1%–0.12%, P = 0.04%–0.045%, S = 0.04%–0.045%, Mn = 0.55%–0.6%.

### 2.2. Test Stand

The columns were subjected to a compression test in the environmental chamber (model 3119-605, INSTRON, Norwood, MA, USA) by increasing the temperature on the stand equipped with the INSTRON machine—model 5982 (INSTRON) (Figure 2). This machine ensures performing tests for compression and tension with a range of 0.02 N to 100 kN. The applied thermal chamber enables raising the temperature inside up to 350 °C. For recording the deformations of studied samples, a non-contact DICAS [24] composed of two tracking cameras was used. Before the test, the specimens were sprayed with the paint resistant to high temperature. To warm up the specimens up to the appropriate temperature in the entire volume, the measurement of the temperature and the load took place every 40–60 min (the data were registered when the reaction force remained constant).

### 2.3. FE Model

Numerical simulations were performed based on the finite element methodby the NASTRAN/MARC FEA 2010R^®^ version software in reference [25]. By using first-order solid finite elements, discrete models were divided, as shown in Figure 3. The generation of finite elements of the column models was achieved by extruding shell elements using a sweep operation. Across the thickness of the column, the number of elements amounted to 4. The total number of finite elements for the discrete model of the column was 163,455. The number of tetragonal elements for each plate amounted to about 50k. The computation was done by applying Green–Lagrangian equations for large strains and deflections. The number of substeps for the single variant was assumed to be from 50 up to 10,000. The maximum number of possible iterations during each substep was set to 500. The simulation was composed of two stages (steps) as was executed in the experiment. In the first step (BC_1), small displacements (*U_y_*) of plates were initiated to close the gaps between the grooves and the edges of the column and to prestress the samples (see Figure 4). Between the column and the plates (on deformable bodies), the contact elements were imposed. None of the friction parameters for contact analysis were taken into account. The contact detection parameter was set to 0.9 (bias on tolerance). The separation criterion was based on forces.

The second stage of the calculations included the thermal analysis by increasing the column temperature until the maximum load was achieved. During the second analysis, both plates on their internal surface were fully constrained (*U_x_* = *U_y_* = *U_z_* = 0). The boundary conditions were inserted in Table 1. In all models, imperfections of columns were not taken into consideration.

### 2.4. Semi-Analytical Method

For orthotropic rectangular plates subjected to thermal-mechanical loads, physical relations combining stresses, strains, and displacements taking into account temperature rise in the middle plane of i-th plate are described as:(1)$$\begin{array}{l} \varepsilon_{xi}=\frac{\sigma_{xi}}{E_{xi}}-\nu_{yxi}\frac{\sigma_{yi}}{E_{yi}}+\alpha_{xi}\Delta T_{0i}=u_{i,x}+0.5(u_{i,x}^{2}+v_{i,x}^{2}+w_{i,x}^{2})\\ \varepsilon_{yi}=\frac{\sigma_{yi}}{E_{yi}}-\nu_{xyi}\frac{\sigma_{xi}}{E_{xi}}+\alpha_{yi}\Delta T_{0i}=u_{i,y}+0.5(u_{i,y}^{2}+v_{i,y}^{2}+w_{i,y}^{2})\\ \gamma_{xyi}=\frac{\tau_{xyi}}{G_{i}}=u_{i,y}+v_{i,x}+u_{i,x}u_{i,y}+v_{i,x}v_{i,y}+w_{i,x}w_{i,y} \end{array}$$
where *σ**_xi_*, *σ**_yi_* denote membrane stresses in x- and y-direction, respectively. *E_xi_* and *E_yi_* represent elastic moduli in x- and y-direction, respectively. Coefficients *ν**_xyi_* and *ν**_yxi_* are Poisson’s ratios where the first index denotes the transverse direction with respect to a direction of the load action, *G_i_* is the shear modulus, *α**_xi_* and *α**_yi_* are thermal expansion coefficients in x- and y-direction, respectively and Δ*T_0i_* is the temperature increment of mid-plane. From the principle of virtual works for a single plate strip, differential equations of equilibrium (2), kinetic and static conditions of a continuum in places of touching strips, and boundary conditions on edges of plate (*x* = 0, *x* = L) can be formulated as given below:(2)$$\begin{array}{l} {\left[N_{1}(1+u_{,1})+N_{3}u_{,2}\right]}_{,1}+{\left[N_{2}u_{,2}+N_{3}(1+u_{,1})\right]}_{,2}=\;0\\ {\left[N_{1}v_{,1}+N_{3}(1+v_{,2})\right]}_{,1}+{\left[N_{2}(1+v_{,2})+N_{3}v_{,1}\right]}_{,2}=0\\ {\left(tM_{1,1}+N_{1}w_{,1}+N_{3}w_{,2}\right)}_{,1}+{\left(tM_{2,2}+2tM_{3,1}+N_{2}w_{,2}+N_{3}w_{,1}\right)}_{,2}=0 \end{array}$$

The problem of the stability was solved by using asymptotic Koiter’s method in reference [23]. The displacement fields (U) and internal forces (N) were developed in exponential series regarding linear amplitude of eigen-buckling *ξ* (determined by the condition of maximum deflection corresponding to the thickness of the first plate *t*_1_)
(3)$$\begin{array}{l} U=\lambda U^{\left(0\right)}+\xi_{n}U^{\left(n\right)}+\cdots \\ N=\lambda N^{\left(0\right)}+\xi_{n}N^{\left(n\right)}+\cdots \end{array}$$
where: U^(0)^ and N^(0)^ stand for displacements fields and cross-section forces fields for a pre-critical state, U^(n)^ and N^(n)^ stand for displacements fields and cross-section forces fields for n-th of buckling mode. By altering equations (3) to equilibrium equations (2), continuity conditions and boundary conditions, the boundary problem of zero and the first order were obtained as in reference [26]. Zero approximation describes the pre-critical state, whereas the first approximation order allows determining critical loads and buckling modes by accounting for the minimization with regard to half-waves number *m* in the longitudinal direction. The program based on fundamental equations presented in work [27] was modified by adding thermal terms. The system of first-order differential equations was solved by the method of the matrix shift, integrating numerical equilibrium equations in the circumferential direction to obtain relations between state vectors on both longitudinal edges by using Gudnov orthogonalization method.

### 2.5. Material Data

Due to one-directional tension tests, full material characteristics at the ambient temperature were determined (Figure 5a). Obtained curves were transformed to the true relation of stress-strain to export to the numerical program (Figure 5b). The average Young’s modulus and Poisson’s ratio were defined as 160 GPa and 0.3, respectively. Regarding the range of temperature affecting samples, Poisson’s ratio and thermal expansion coefficient were assumed to be independent of temperature and were equal to 0.3 and α = α_xi_ = α_yi_ = 10^−5^ 1/K, respectively.

## 3. Results

### 3.1. Buckling Forces

This section was devoted to the analysis of critical loads. To refer to the experiment, the buckling forces ($F_{cr}^{SAM}$, $F_{cr}^{FEM}$), the buckling temperature increments ($\Delta T_{cr}^{SAM}$, $\Delta T_{cr}^{FEM}$) and then the relating reaction forces ($F_{Rcr}^{SAM}$, $F_{Rcr}^{FEM}$) of the articulated supports of the column on its edges were determined using FEM and SAM. Critical reaction forces were calculated by multiplying critical temperatures, Young’s modulus, the cross-section area of the column, and the thermal expansion coefficient (formula taken from Hook’s law). The assumed type of boundary conditions in calculations of buckling loads differs slightly from that applied in the experiment, but in the case of the experimental study, it was hard to realize adequate supports (shape of grooves) in the middle of each edge. In the case of the determination of the critical temperatures, the uniform increment of the temperature in the entire structure of the column was considered. Of course, both methods gave comparable results in terms of the critical forces and the critical temperatures (Table 2). Accounting for SAM, the critical forces and the critical reaction forces caused by the temperature have the same values. Referring to FEM results, differences between them are about 1% (compression produces light shortening, but the temperature rise causes a reverse effect). Buckling mode maps relating to the critical loads (forces and temperatures) achieved numerically are presented in Table 3. As can be seen, the modes of detailed loads are almost the same (numbers of half-waves along the column axis are the same).

### 3.2. Analysis of Column Compression

To better understand the behavior of the columns, curves of load F vs. shortening Δ have been illustrated in Figure 6 for all considered analysis methods. Based on analytical results (SAM), the maximum load of the column is not observed with reference to the assumption of elastic material in the whole range. The achieved load carrying capacity of the compressed column in FEM corresponds closely to the maximum load recorded in the experiment (values taken from three attempts). The noticed difference in the maximum loads amounts to a few percent ($F_{\max}^{FEM}$ = 14.48 kN and $F_{\max}^{EXP}$ = 14.22 kN). The discrepancies between curves are seen with respect to full shortening of the column. In the case of FEM, the range of shortening is considerably smaller than in the experiment (about two times).

### 3.3. Analysis of Thermal Compression

This section concerns the analysis of the full path of the post-buckling state of the columns. Based on the FEM results, Figure 7 depicts the influence of the initial forces *F_int_* (preloads) on the maximum temperature increment until the force decreases. With the small initial load (*F_int_* = 0.68–2.36 kN), at the beginning of the temperature increase, the compressing reaction force grows slowly (due to the adjustment of the specimen in grooves).

For higher initial loads, the proportional characteristics are seen but only in a limited range. Moreover, the maximum registered loads (load carrying capacity) are almost the same values ($F_{R\max}^{FEM}$ = 14.49–14.62 kN) independent of the initial forces before heating the columns. These small differences may be due to the application of numerical methods.

The trend of curves is repeatable, but after exceeding first critical loads (given in Section 3.1—$F_{cr}^{SAM}$ = 10.72 kN and $F_{cr}^{FEM}$ = 10.82 kN), temperatures rise with a slight growth of the reaction force. Figure 8 presents these same curves obtained by the experiment (EXP-experimental results). The tests were conducted for a few initial preloads (curves for five variants were obtained). For each considered load case (variants), three tests were executed, but average values of reaction forces regarding their adequate temperature increment are plotted in Figure 8 and Figure 9.

In the case of the experiment, the curves have similar trends as in FEM, but the received maximum loads and the maximum temperature increment differ slightly from each other. These characteristic values are shown in Table 4. To highlight differences, some curves obtained experimentally and numerically are compared and plotted in Figure 9. Taking into consideration the maximum reaction forces obtained numerically and experimentally, the discrepancy amounted to a few percent (for F_int_ = 1.35 kN–5.2%, for F_int_ = 2.0 kN–3.5%, for F_int_ = 3.0 kN–3.7%, F_int_ = 4.0 kN–0.1%, and for F_int_ = 8.0 kN–4.53%). To better visualize the behavior of the columns in the thermal environment, the deformation maps of the specimens based on DICAS and FEM are shown in Table 5. Many points of the measurements gave quite a good correlation (for example, cases: F_int_ = 2 kN–point 3, F_int_ = 3 kN–point 3, and F_int_ = 4 kN–point 3).

## 4. Summary

This paper reports the analysis of the post-buckling state of the columns under compression and the compression due to the temperature rise. Turning now to the comparison between the experimental and numerical evidence, two methods of calculation were employed and related to the experiment score: FEM and SAM based on Koiter’s asymptotic theory. The findings of that comparison clearly indicate that:SAM in relation to FEM gives comparable critical loads and temperatures of the column with the articulated supports on the edges;The obtained compression curve of SAM in comparison to FEM run close to each other, but in comparison to the experiment, the total shortening of the columns at the damage point is two times smaller. However, the maximum loads of the temperature increment differ from each other moderately for the small initial loads (See Table 4);Referring to the curves of FEM and EXP, it is observed that a good agreement was achieved especially for the higher initial preloads (see Figure 9 and Table 4). The large discrepancy in characteristics for the minimal initial loads may result from the imperfect specimens and accuracy of the whole set-up;Based on the records obtained by DICAS and FEM, the modes of deformation at the thermal loads are similar. Moreover, values of displacements on the surface of the columns are comparable;The large deformations of the column during its failure followed in the vicinity of the one of support places (see Table 5, point 3 for all variants);Comparing the compression of the column due to the grips’ approach or the temperature growth, maximum load mostly remains on the same level.

## Figures and Tables

**Figure 1 materials-13-00074-f001:**
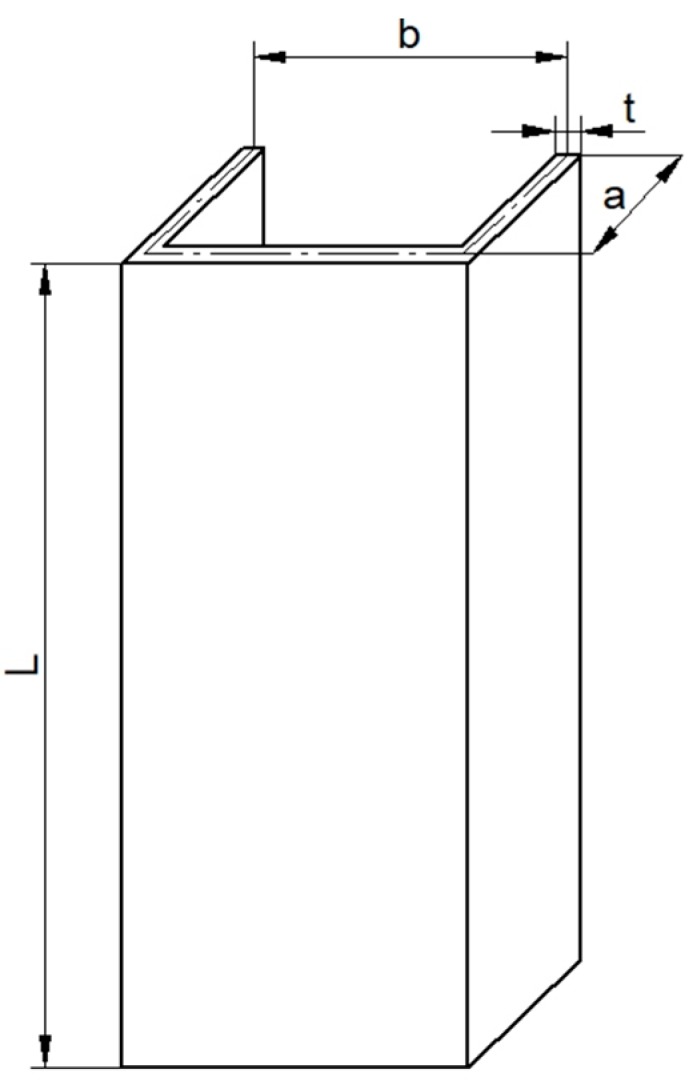
The object of the study.

**Figure 2 materials-13-00074-f002:**
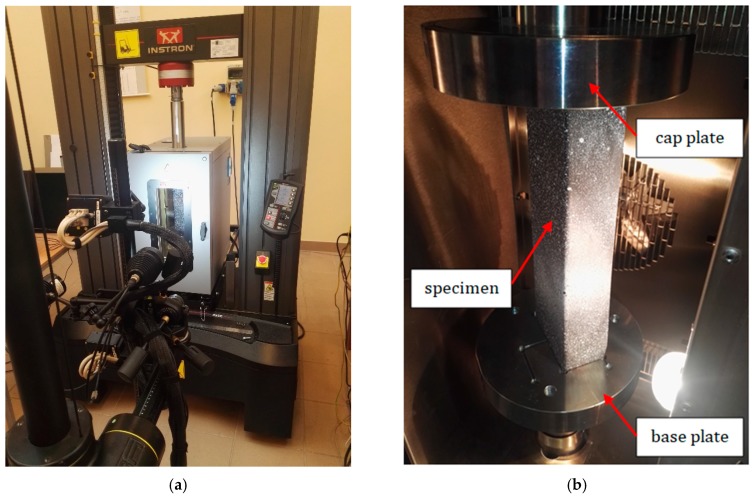
The stand with Aramis system (**a**) and the column in the chamber before the test (**b**).

**Figure 3 materials-13-00074-f003:**
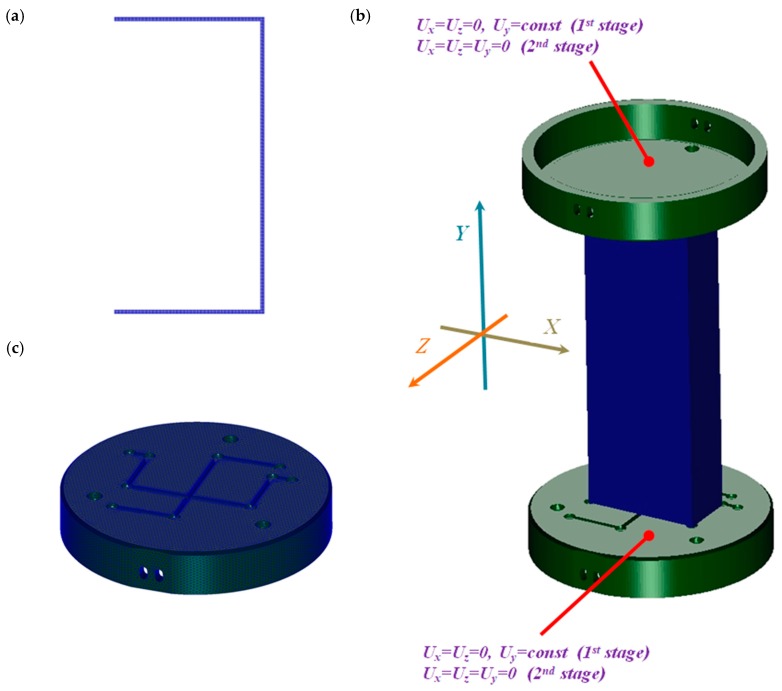
The finite element (FE) model of the column (**a**) and the discrete model of the plates (**b**) and the whole model with the boundary conditions (**c**).

**Figure 4 materials-13-00074-f004:**
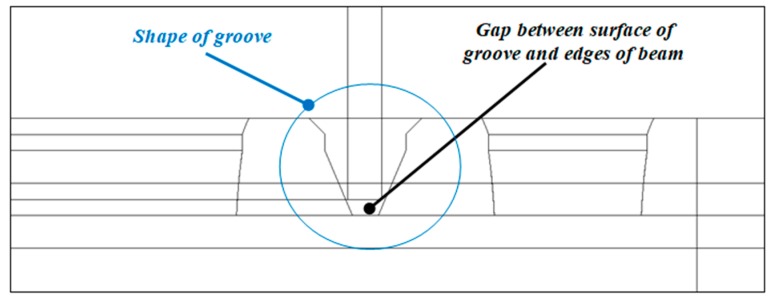
The view of the groove shape in relation to the column edges.

**Figure 5 materials-13-00074-f005:**
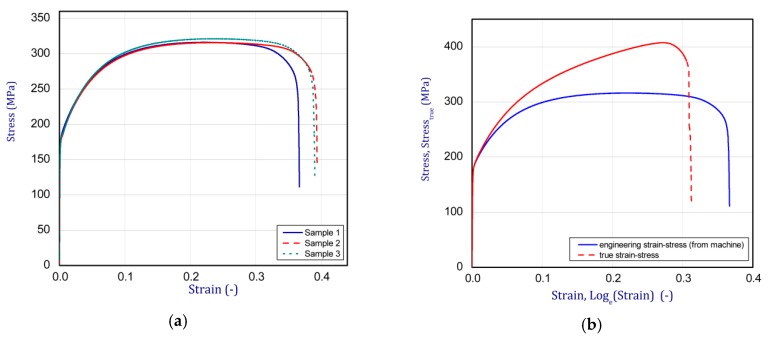
The tensile test characteristics of cold-rolled steel (DC01) (**a**) and transformation into true strain-stress relation (**b**).

**Figure 6 materials-13-00074-f006:**
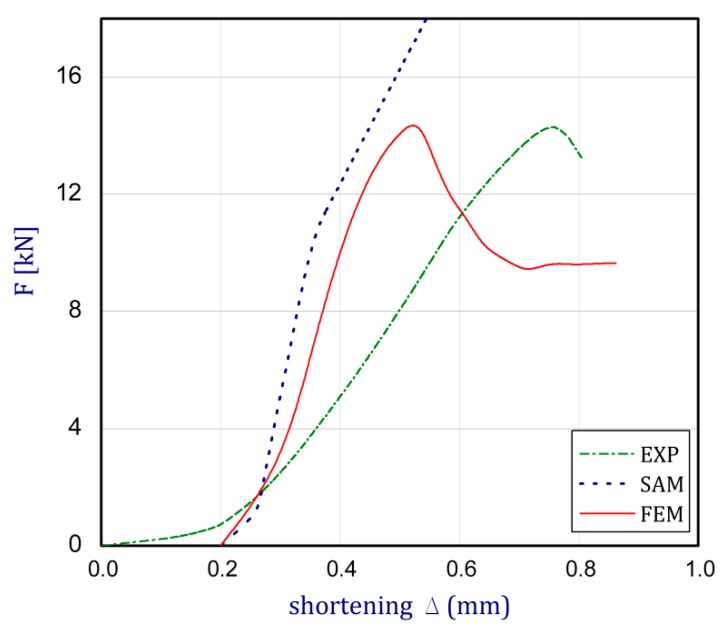
The diagram of the column compression (shortening vs. compressed force).

**Figure 7 materials-13-00074-f007:**
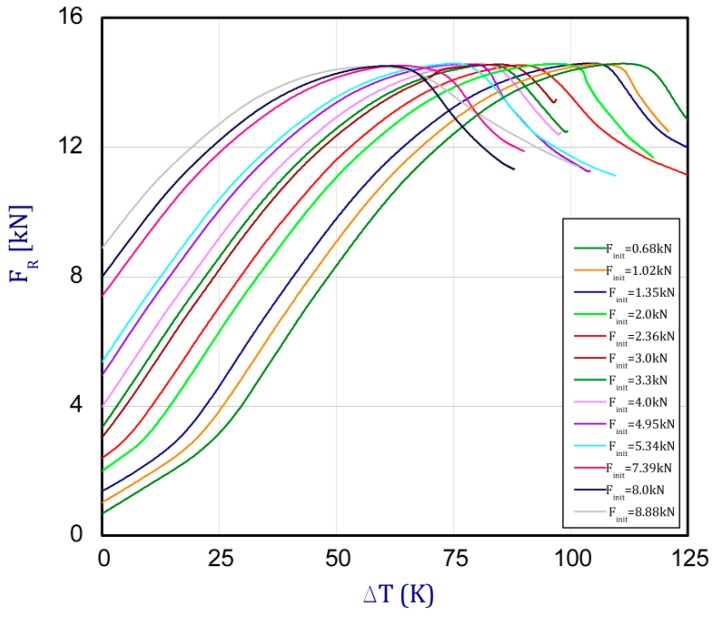
The force reaction vs. the temperature increment for different preloads obtained by FEM.

**Figure 8 materials-13-00074-f008:**
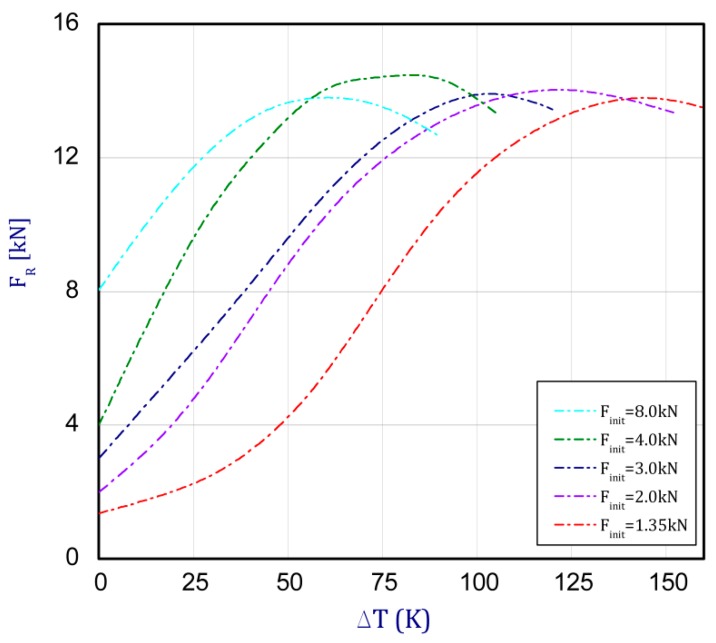
The force reaction vs. the temperature increment for different preloads obtained by EXP.

**Figure 9 materials-13-00074-f009:**
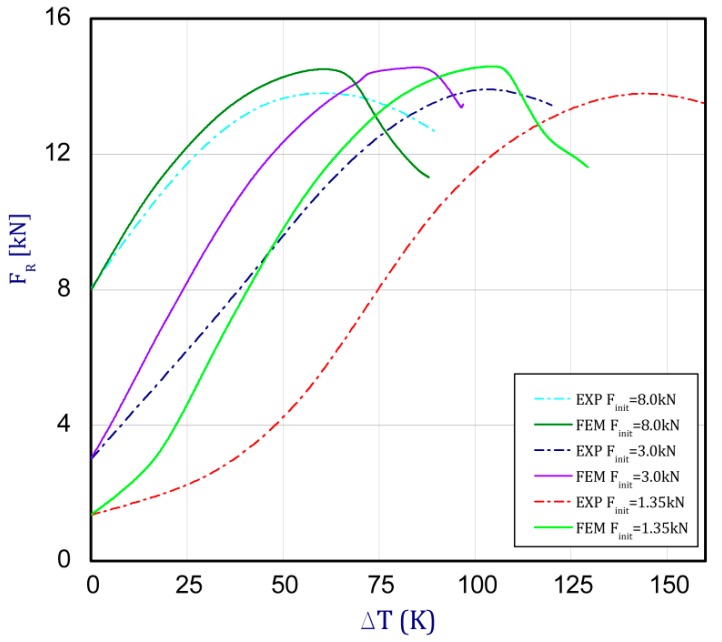
Comparison of obtained curved by FEM and EXP.

**Table 1 materials-13-00074-t001:** The boundary conditions of the internal surface of the plates.

Boundary Condition Number	U_x_	U_y_	U_z_	ΔT
BC_1	0	const	0	–
BC_2	0	0	0	const

**Table 2 materials-13-00074-t002:** The critical loads obtained by using two methods.

Mode	Critical Forces SAM $F_{cr}^{SAM}\;(\mathrm{kN})$	Critical Forces FEM $F_{cr}^{FEM}\;(\mathrm{kN})$	Critical Temperature (Reaction Forces) SAM $\Delta T_{cr}^{SAM}\;(K)$ $\left(F_{Rcr}^{SAM})\;(\mathrm{kN}\right)$	Critical Temperature (Reaction Forces) FEM $\Delta T_{cr}^{FEM}\;(K)$ $\left(F_{Rcr}^{FEM})\;(\mathrm{kN}\right)$
1	10.715	10.977	41.86 (10.71)	42.26 (10.81)
2	11.010	11.309	43.01 (11.01)	43.59 (11.16)
3	13.015	13.373	50.84 (13.01)	51.73 (13.243)
4	13.729	14.329	53.63 (13.73)	55.55 (14.221)
5	14.748	15.524	57.61 (14.75)	60.12 (15.391)

**Table 3 materials-13-00074-t003:** The buckling modes of the columns.

Type of Buckling	Number of Modes				
	1	2	3	4	5
Due to temperature	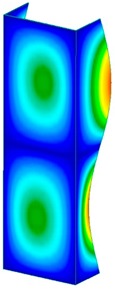	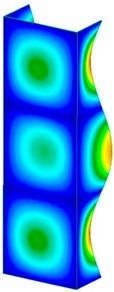	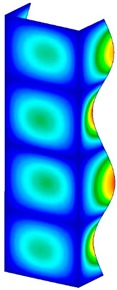	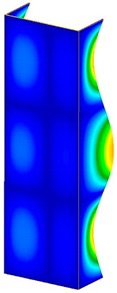	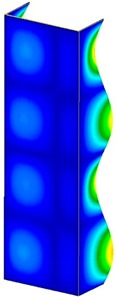
Due to compression	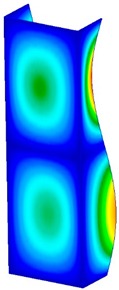	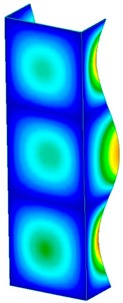	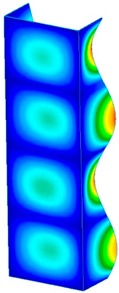	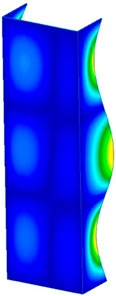	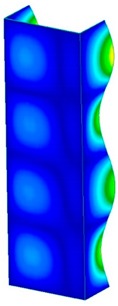

**Table 4 materials-13-00074-t004:** The maximum values of the temperature increment and the reaction forces.

F_int_ (kN)	Maximum Temperature Increment EXP $\Delta T_{\max}^{EXP}\;(K)$	Maximum Temperature Increment FEM $\Delta T_{\max}^{FEM}\;(K)$	Maximum Load EXP $F_{R\max}^{EXP}\;(\mathrm{kN})$	Maximum Load FEM $F_{R\max}^{FEM}\;(\mathrm{kN})$
0.68	nd	113.38	nd	14.61
1.02	nd	107.38	nd	14.58
1.35	148	105.17	13.87	14.61
2.00	122	98.28	14.11	14.59
2.36	nd	92.40	nd	14.60
3.00	105	89.21	14.08	14.59
3.30	nd	86.16	nd	14.61
4.00	90	85.02	14.61	14.60
4.95	nd	81.25	nd	14.57
5.34	nd	79.23	nd	14.62
7.39	nd	68.88	nd	14.56
8.00	61	63.20	13.89	14.52
8.88	nd	56.28	nd	14.49

**Table 5 materials-13-00074-t005:** The deformation maps (total displacements) based on the Digital Image Correlation ARMIS® system (DICAS) and FEM.

Point	*F_int_* = 2 kN		*F_int_* = 3 kN		*F_int_* = 4 kN	
	FEM	DICAS	FEM	DICAS	FEM	DICAS
1	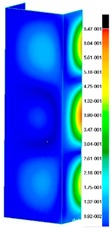	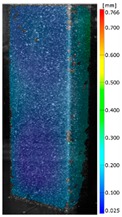	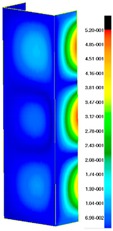	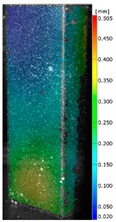	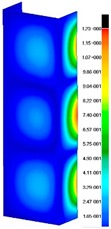	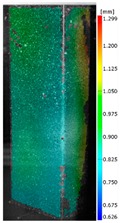
**Values**	Δ*T* = 34 K *F_R_* = 8.08 kN	Δ*T* = 38 K *F_R_* = 7.85 kN	Δ*T* = 38 K *F_R_* = 6.46 kN	Δ*T* = 42 K *F_R_* = 6.32 kN	Δ*T* = 29 K *F_R_* = 9.20 kN	Δ*T* = 30 K *F_R_* = 8.05 kN
2	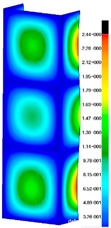	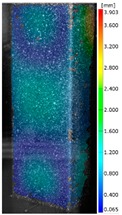	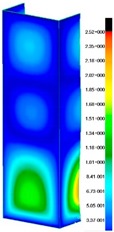	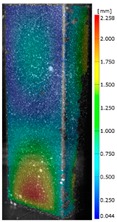	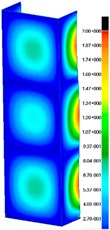	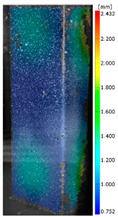
**Values**	Δ*T* = 92 K *F_R_* = 14.50 kN	Δ*T* = 96 K *F_R_* = 14.20 kN	Δ*T* = 87 K *F_R_* = 13.64 kN	Δ*T* = 90 K *F_R_* = 12.80 kN	Δ*T* = 60.5 K *F_R_* = 12.60 kN	Δ*T* = 61 K *F_R_* = 12.26 kN
3	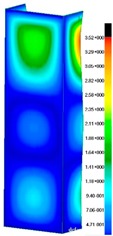	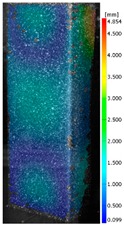	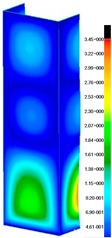	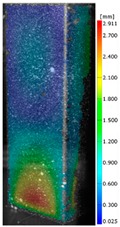	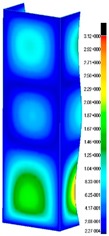	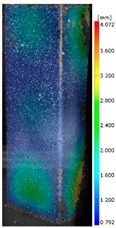
**values**	Δ*T* = 112 K *F_R_* = 12.30 kN	Δ*T* = 110 K *F_R_* = 14.09 kN	Δ*T* = 98 K *F_R_* = 12.20 kN	Δ*T* = 102 K *F_R_* = 12.50 kN	Δ*T* = 99 K *F_R_* = 12.12 kN	Δ*T* = 98 K *F_R_* = 12.01 kN

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
