# Peer review of "The Buckling and Post-Buckling of Steel C-Columns in Elevated Temperature"

_materials, 2019, doi:10.3390/ma13010074_

Round 1

Reviewer 1 Report

The manuscript is overall well written, the experiments are described in detail and the English language is adequate. However, in terms of the studied material and the applied experimental methods no significant novelty is presented (they seem rather common).   One aspect should be more clearly specified: the number of samples used for the compression tests. From figure 5a one could conclude that 3 samples were tested. Of course that there were multiple tests performed with different parameters. However, the number of tests performed on the same type of sample and with the same parameters is not specified. Therefore some questions regarding the experiments reproducibility could arise, since the standard deviation it is not discussed or presented in the manuscript.   To conclude, the article can be published after minor revision IF the editors consider that the subject of the manuscript is of interest to the journal readers.

Author Response

Dear Reviewer,

first of all, we’d like to thank for your valuable remarks, which let us improve our paper. Taking into consideration these suggestions, we have done our best to refine our article and to fulfil the requirements of publishable standard. The added/changed text was highlighted on “yellow”.

Best Regards,

Authors of manuscript

Reviewer 2 Report

A minor revision is given to this paper before it can be published. My main opinions are here:

The authors need to rewrite abstract in order to make it clearer to the reader. The originality and the main results should be included. The authors should revise the paragraph 1 by adding the experiment results in the corresponding references and removing a couple of references. Adding the corresponding references after the sentence “The edges of columns during tests were inserted into grooves of the plates to reflect articulated supports often used in the literature”. The authors should explain the standard to set thermal load in section 2.2. The authors should revise some of figures such as figure no. 7, 8 and 9. For example, solid with different colors represents the results obtained by FEM and dotted with different colors represents the results obtained by EXP. The references format should be uniform. For example, the journal title should be either full or abbreviation.

Author Response

(The authors gave the same response as above.)
